# How Functional Connectivity Measures Affect the Outcomes of Global Neuronal Network Characteristics in Patients with Schizophrenia Compared to Healthy Controls

**DOI:** 10.3390/brainsci13010138

**Published:** 2023-01-13

**Authors:** Kamil Jonak, Magdalena Marchewka, Arkadiusz Podkowiński, Agata Siejka, Małgorzata Plechawska-Wójcik, Robert Karpiński, Paweł Krukow

**Affiliations:** 1Department of Computer Science, Lublin University of Technology, 20-618 Lublin, Poland; 2Department of Clinical Neuropsychiatry, Medical University of Lublin, 20-059 Lublin, Poland; 3Mechanical Engineering Faculty, Lublin University of Technology, 20-618 Lublin, Poland; 4Da Vinci NeuroClinic, Lublin, 20-102 Lublin, Poland; 5Faculty of Medicine, Medical University of Lublin, 20-059 Lublin, Poland; 6Department of Machine Design and Mechatronics, Faculty of Mechanical Engineering, Lublin University of Technology, 20-618 Lublin, Poland; 7Chair and I Department of Psychiatry, Psychotherapy, and Early Intervention, Medical University of Lublin, 20-059 Lublin, Poland

**Keywords:** functional connectivity, schizophrenia, EEG, neuronal networks, *PLI*, *PLV*, MST

## Abstract

Modern computational solutions used in the reconstruction of the global neuronal network arrangement seem to be particularly valuable for research on neuronal disconnection in schizophrenia. However, the vast number of algorithms used in these analyses may be an uncontrolled source of result inconsistency. Our study aimed to verify to what extent the characteristics of the global network organization in schizophrenia depend on the inclusion of a given type of functional connectivity measure. Resting-state EEG recordings from schizophrenia patients and healthy controls were collected. Based on these data, two identical procedures of graph-theory-based network arrangements were computed twice using two different functional connectivity measures (phase lag index, *PLI*, and phase locking value, *PLV*). Two series of between-group comparisons regarding global network parameters calculated on the basis of *PLI* or *PLV* gave contradictory results. In many cases, the values of a given network index based on *PLI* were higher in the patients, and the results based on *PLV* were lower in the patients than in the controls. Additionally, selected network measures were significantly different within the patient group when calculated from *PLI* or *PLV*. Our analysis shows that the selection of FC measures significantly affects the parameters of graph-theory-based neuronal network organization and might be an important source of disagreement in network studies on schizophrenia.

## 1. Introduction

Psychiatric diseases and milder forms of psychological health disturbances have become one of the key social and health challenges around the world. According to a recent report on the prevalence of mental health problems in the US (2020), in 2019 about 51.5 million American adults (20.7% of the population) suffered from some kind of mental illness, and 13.1 million adults were diagnosed with serious mental illness (SMI), which mainly includes severe diseases that significantly hinder or prevent independent functioning, education, and paid work. In this context, it is of vital importance that in both mentioned populations young adults aged 18–25 had the highest age-related prevalence of psychological health problems [1]. Schizophrenia is one of the most burdening mental illnesses with a clinical onset occurring in adolescence or early adulthood. The disease development commonly has a slow and hidden course with a long-lasting increment of so-called negative syndromes, i.e., affective blunting, social isolation, and loss of interests and initiative. The second phase usually relates to the exacerbation of more or less explicit psychotic symptoms such as auditory verbal hallucinations and delusions, which might be described as incorrect judgments of reality and other people’s behavior, i.e., in the form of delusions of persecution or grandeur [2,3]. This well-known psychiatric psychopathology is accompanied by the presence of cognitive dysfunctions affecting patients’ abilities to comprehend complex language communication, maintain focused attention, plan, solve everyday problems, and generally regulate goal-directed behaviors [4,5].

Despite the existence of numerous theoretical approaches to understanding the etiology of schizophrenia, one may say that currently a substantial group of neuroscientists acknowledge schizophrenia as a brain connectivity disorder, given the vast body of evidence documenting neuronal miswiring and disturbances in the organization of functional integration at the levels of synapses, groups of neurons, hemispheres, and the so-called large-scale neuronal networks [6,7]. The disconnection hypothesis suggests that individual symptoms might be explained with reference to abnormalities regarding mechanisms granting the optimal coordination of a given group of neuronal structures. On the other hand, it enables an understanding of the complexity of the psychosis’s clinical picture as a result of multilayered disruptions in the organization of the whole brain and not only damage or hypofunction encompassing individual cortical areas or subcortical structures [8,9].

Undisputed progress in research on schizophrenia conducted with the application of functional connectivity (FC) measures and network theory solutions caused a noticeable advance in the modern comprehension of the disease as a systemic pathology of the nervous system, but on the other hand, the rapid introduction of many computational methods used in the reconstruction of the neural network gave rise to some methodological confusion related to the multiplicity of mathematical algorithms of similar functions used in parallel. This peculiar excess of algorithms and computational designs may be one of the sources of heterogeneity in the obtained results, especially since the very process of network reconstruction is multiphase and involves the use of various signal processing techniques at different stages [10,11,12]. Finding the optimal computational tool to establish the unique features of schizophrenia-related patterns of functional connectivity and global-scale network configuration seems to be an increasingly important goal in clinical neuroscience, especially considering the still unmet goal of elaborating differential diagnosis methods based on objective biological markers [13].

Considering the above, our analyses aimed to verify to what extent the selection of just two different functional connectivity indicators could affect the obtained results regarding the comparison of the structure of the global neural network in a sample of patients diagnosed with schizophrenia compared to a demographically similar control group. Analyses of global neural network configuration carried out with the use of graph theory (our analysis applied the minimum spanning tree [14]) provide many characteristics describing the organization of brain activity, and above all they inform whether a given network is dominated by mechanisms of integration or selection [15] and whether a network processes information according to the principle of reduced wiring costs and efficiency [16]. Network research in schizophrenia, including that using graph theory and its characteristics, such as the path length, clustering coefficient, and small-worldness, is particularly important because it captures the activity of the whole brain as an organized or disorganized system [17]. However, we postulate that the computational complexity of these analyses and the potentially unhampered freedom of choice regarding applied functional connectivity measures may make them susceptible to volatility and eventually result in low outcome reproducibility.

Therefore, we assumed that our study could generate two main findings:(1).Establishing whether and to what extent the graph theory network parameters (e.g., path length) show independence from the input data (FC measures) in terms of the range and direction of difference between patients and healthy controls regarding a given graph parameter;(2).If it proved that selected graph-theory indicators are independent of the exact measure of connectivity strength, it would be possible to determine which FC parameter used as an element of the global network computation differentiates patients and controls to the greatest extent. Indicating such FC measures may guide future research.

By the independence of network parameters from the input data, we mean that the direction of differentiation between patients and controls is the same and that these two groups can be differentiated by the same network parameters, regardless of the FC measure used for their calculation. Moreover, the lack of significant intragroup differences in the graph-theory-based network parameters calculated on the basis of two different FC measures was considered an indicator of the non-susceptibility of these algorithms to input factors. However, if graph-theory parameters, calculated on the basis of two different FC measures, give completely different results in terms of the differences and similarities between the two groups, this indicates that the network analysis outcomes are more dependent on the specificity of the included FC measures and consequently that the selection of a specific FC algorithm creates the final results regarding the specificity of the whole-brain network architecture.

## 2. Materials and Methods

### 2.1. Participants

The study included a group of patients diagnosed with schizophrenia, according to the DSM-5 classification, aged 20–35, with at least 10 years of education. The following exclusion criteria were taken into account: prior diagnosis of an intellectual disability, psychoactive substance addiction, structural abnormalities of the brain or other MRI indicators of its acquired damage (e.g., post-traumatic or vascular changes), comorbidity of neurological diagnoses, taking benzodiazepines and antiepileptic drugs, more than three psychotic episodes requiring hospitalization, and pronounced features of metabolic syndrome. The patients included in the study group had to be treated only with atypical antipsychotics. The patients came from the first Psychiatry Department of the Medical University of Lublin. After assessing the clinical group and determining its basic demographic characteristics, healthy individuals were selected to form a control group using a pairwise selection method. The exclusion criteria in the control group were the diagnosis of mental and neurological diseases and disorders, head injuries and concussions, and taking medications that may affect EEG recordings (e.g., hypnotics and benzodiazepines). Subjects did not receive remuneration for participating in the study. All participants gave their written consent for the study, and the research project was positively assessed by the local ethics committee.

### 2.2. EEG Recording Acquisition

First, for each participant, 15 min of resting-state EEG (eyes closed) data were recorded with 19 scalp position, electro-cap (Electro-Cap International Inc., Eaton, OH, USA), and Ag/AgCl disk electrodes. Electrodes were distributed according to the 10–20 International system (Fp1, Fp2, F3, F4, C3, C4, P3, P4, O1, O2, A1, A2, F7, F8, T3, T4, T5, T6, Fz, Pz, and Cz). During the acquisition, subjects sat in a well-lit and quiet room. The electrode impedances were kept below 5 k, and the data were filtered from 0.5 Hz to 70 Hz (with an active notch filter set to 50 Hz) when the sampling rate was 512 Hz. The data were exported into ASCII format after recording. The post-processing procedure was made in the EEGLAB 2021.0 [18] program, which is a Matlab 2018a toolbox. In the first step of the post-processing procedure, the signal was filtered with a 0.5–45 Hz bandpass filter (second-order Butterworth filter). Second, the reference was changed offline into the average one. Third, from the processed signal, 75 eight-second-long epochs (4096 samples) without artifacts were extracted for each patient by a clinical neurophysiologist. Lastly, the EEG signals were divided into six frequency bands using finite impulse response filters: delta (0.5–4 Hz), theta (4–8 Hz), low alpha (8–10 Hz), high alpha (10–12 Hz), beta (13–30 Hz), and gamma (30–45 Hz).

### 2.3. Functional Connectivity Indicators: PLI and PLV

The first measure used to compute the functional connectivity strength was the phase lag index (PLI). The *PLI* is a phase synchronization metric based on the asymmetry of the distribution of phase differences between two signals, which may be calculated using the analytical signal based on the Hilbert transform. When compared to other synchronization detection approaches, such as synchronization likelihood or phase coherence, the *PLI* is less influenced by the shared source because zero-lag synchronization has been excluded from the analysis. According to Stam and coworkers, “The phase lag index is based upon the idea that the existence of consistent, nonzero phase lag between two time series cannot be explained by volume conduction from a single strong source and therefore, renders true interactions between the underlying systems rather likely” [19] (p. 1179). The *PLI* is obtained from the phase difference ($\Delta \phi (t_{k}),k=1\ldots N$) of the time series by means of
$$PLI=\left|\langle sign[\Delta \phi (t_{k})]\rangle \right|$$
where Δ*ϕ* is the phase difference and <…> denotes the average time (*t*). The *PLI* quantifies the relative phase distribution’s asymmetry, which indicates that the likelihood that the phase difference (Δ*ϕ*) will be in the interval −*π* < Δ*ϕ* < 0 is different from the likelihood that it will be in the interval 0 < Δ*ϕ* < *π*. The range of *PLI* values is between 0 and 1, where a zero value indicates no coupling or coupling with the phase difference centered around 0 (mod *π*) and 1 values indicate perfect phase locking at a value of Δ*ϕ* different from 0 (mod *π*). The stronger nonzero phase locking means that the *PLI* is larger [19].

The second FC measure was phase locking value (PLV) [20]. The *PLV* quantifies similar synchronization tendencies in EEG signals. The advantage of *PLV* is that it can compute the phase component separate from the amplitude component for a particular frequency range. The *PLV* assesses the latencies at which phase synchrony or modest phase variation occur across trials in situations with recurrent stimuli. The *PLV* computing process incorporates the instantaneous phase difference between signals in the chosen frequency band in order to establish the phase synchronization of two EEG signals. The synchronization measure *PLV*, at time instant *t* is defined as:$$PLV=\frac{1}{N\;}\left|\sum_{n-1}^{N}e^{j\left(\Delta \phi \left(t,n\right)\right)}\right|$$
where *N* is the total number of trials and Δϕ(t,n) = ϕ1(t,n)−ϕ2(t,n) is the instantaneous phase difference between the signals.

The *PLV* is used in the majority of EEG research to assess the intertrial variability in phase at time *t*. The *PLV* is close to one if there is no substantial phase change between trials. A *PLV* value of zero indicates that the phase difference between the two signals is not synced, while a *PLV* value of one shows that the signals are totally synchronized [21].

### 2.4. Global Neuronal Network Reconstruction: Application of the Minimum Spanning Tree

The functional connectivity matrixes created by the *PLI* and *PLV* for each frequency band were converted into graphs. Every graph consisted of nodes (i.e., EEG electrodes) and edges (i.e., functional connectivity values between each pair of electrodes gathered from *PLI*/*PLV*). By applying Kruskal’s algorithm to each *PLI*/*PLV* adjacency matrix, calculated for each frequency band epoch acquired from each participant, a minimum spanning tree (MST) graph was constructed. The first step in this process was to sort the edge weights from smallest to largest. Once all nodes were separated, the algorithm began reconnecting them, starting with the node that had the greatest weight. The algorithm then continued to add the connection with the next greatest weight until all of the nodes were linked. In contrast, if a new connection with a node during the adding method caused a cycle or loop, the connection was refused, and the next edge was rated by the weight value [15]. Various graph metrics could be established throughout the MST computation process. However, in order to keep the computational methodology in line with the original MST basis, we generated results describing eight parameters presented in the official guide and MST computation instructions developed by Cornelius J. Stam’s team [22]. In accordance with the work of van Dellen and coworkers [14], the most straightforward MST parameters are the diameter and leaf number/leaf fraction. Furthermore, these variables enable the classification of networks as integration- or segregation-dominated. When a network topology expresses an increased diameter and a decreased leaf number, it is a so-called line-like network topology. The second network type has a low diameter and a high leaf number, called a star-like network, in which integrative processes dominate [15]. The leaf fraction can be defined as the number of nodes on a tree with degree = 1 and can be calculated from: Lf = L/m, where L is the leaf number. The leaf number range starts from two (typical of a line-like topology), and its maximum value is equal to m = N1 (a star-like topology). The leaf number is associated with the tree diameter (d) parameter, which can be defined as the largest distance between any two nodes. Additionally, apart from those mentioned global MST metrics, one can also describe the optimal tree topology with a function called tree hierarchy. Tree hierarchy pictures the transfer of information from one node to another in the shortest path, assuming that there is no overload in the central node of the tree. Additional MST metrics might be evaluated, and these are kappa, R, Teff, ASP and the mean. Parameter R is a derivative of the Pearson correlation coefficient, indicating the assortativity level, i.e., the feature of the network consisting of the fact that high-degree nodes should be connected with nodes with the same magnitude. However, in more chaotically constructed networks, high-degree nodes can be directly connected with low-degree nodes. The assortativity ranges from −1 to +1, and negative values are more typical in networks where the magnitudes of connected nodes are substantially different [23]. Kappa, also called the degree of divergence, measures the broadness of degree distribution. A decreased value of kappa indicates a decreased number of highly connected nodes called “hubs” [14,15,23]. Teff is defined as: 1 − diameter/(*N* − leaf number + 1). This measure indicates how close the diameter, for a given *N* and leaf number, is to its lowest possible value. Teff ranges between 0 and 1. ASP is an abbreviation meaning the average shortest path computed for the whole MST. This is not a normalized index. Lastly, the mean of the MST is the value of the mean weight of all the edges constituting the MST graph.

### 2.5. Statistical Analyses

Demographic variables were compared between groups with Student’s *t*-test, and nominal variables (e.g., sex) were compared with the *χ*^2^ test. Due to the fact that the subjects from the two groups were matched in pairs and the potential differences in variables such as age, gender, and education were controlled, the comparison of MST parameters calculated on the basis of both the *PLI* and *PLV* between the two groups was carried out using a two-tailed *t*-test. Cohen’s *d* was applied as an effect size indicator. In the intragroup analysis of SCH patients, a within-subjects ANOVA was used, where the above parameters were summarized in two different versions, one based on the *PLI* and the other based on the *PLV*. The partial eta square (*η*_p_^2^) was used as an indicator of the effect size.

## 3. Results

### 3.1. Demographic and Clinical Characteristics of the Studied Groups

After applying inclusion and exclusion criteria, the collected groups consisted of 20 patients with first-episode schizophrenia, including 10 women and 10 men with an average age of 20.20 years. The healthy control groups also consisted of 10 men and 10 women with an average age of 20.10 years. As shown in Table 1, the groups did not differ in terms of age, sex, or education.

All SCH patients were treated with atypical antipsychotics, the majority of them (60%) with Olanzapine; the average dose expressed in risperidone equivalents reached 4.82 ± 0.93 units. According to the exclusion criteria, none of the patients were taking benzodiazepines or anticonvulsants. Four patients (20%) were additionally treated with SSRI antidepressants (sertraline), and nine (45%) were taking chlorprothixene for sporadic insomnia at a single dose not exceeding 30 mg. The duration of illness in the clinical group was about 11 months, and the duration of untreated psychosis was about 4 months.

### 3.2. Between-Group Comparison of MST Outcomes Calculated on the Basis of PLI and PLV

Since SCH patients and individuals from the HC group did not differ in terms of demographic variables, comparisons of MST-related network measures based on *PLI* or *PLV* indicators of functional connectivity were computed directly with the application of a two-tailed Student’s *t* test. Table A1 in Appendix A presents all detailed results of this comparisons with Cohen’s *d* used as an effect size indicator.

Overall, in the case of 23 variables, the between-group difference reached the threshold of *p* < 0.05. Figure 1 presents distribution plots of all statistically significant results that survived FDR correction for repeated testing. Among these variables, 15 were based on the *PLV* and 8 were based on the *PLI*. Considering the effect size variable, the “Mean” differentiated the groups to the greatest extent. The largest intergroup differences concerned the following variables: the *PLV*-based mean MST in the delta frequency (*d* = 4.69), the *PLV*-based mean in gamma frequency (*d* = 4.13), the *PLV*-based mean in the beta frequency (*d* = 3.33), the *PLV*-based mean in the low-alpha frequency (*d* = 2.83), the *PLV*-based mean in the high-alpha frequency (*d* = 2.78), and the *PLI*-based mean in gamma frequency (*d* = 2.35). It is worth noting that in some pairs of MST results the values based on the *PLV* were significantly higher in SCH patients than in controls, while the same MST variables based on the *PLI* in the same frequency presented the opposite difference, i.e., values were lower in SCH patients than in controls. For example, in the gamma band, the mean MST values based on the *PLI* in groups (SCH vs. HC) were 0.144 vs. 0.221, while the values for the *PLV*-based measure were 0.842 vs. 0.553. Besides the MST mean, the other network metrics that differentiated the groups the most (with *d* > 1) were the *PLI*-based ASP (*d* = 1.28), the *PLI*-based diameter (*d* = 1.20), and the *PLI*-based leaf (1.19)—all in the gamma band—and the *PLV*-based kappa in the low-alpha band (*d* = 1.11). As in the case of the MST mean, for some of these variables the opposite directions of difference in measures could also be observed based on the *PLI* and *PLV* indicators (e.g., the ASP in the gamma band).

### 3.3. Within-Group Comparisons of PLI and PLV-Based MST Indicators

Another step of the analysis considered within-group evaluations regarding possible discrepancies between MST variables based on the *PLI* and *PLV* indexes of connectivity strength in SCH patients. We performed within-group ANOVAs separately for each frequency (MST indicators as dependent variables and *PLI* vs. *PLV* as an independent variable) with Bonferroni post hoc test significance level corrections for multiple testing. According to the diagrams displayed in Figure 2, in the majority of cases MST indicators such as the kappa, ASP, and mean were significantly different depending on whether they were calculated on the basis of *PLI* or *PLV* markers, although all came from the same SCH individuals. The mentioned differences reached the level of significance after correction for the number of tests. The interaction effects (FC indicator x MST value) at all analyzed frequencies were also significant (*p* < 0.0001, *η*_p_^2^ > 0.4) after correction for multiple comparisons. In detail, within the SCH group, there were significant differences at all six analyzed frequencies regarding the mentioned indicators. Additionally, in the delta band, the R was also significantly different for computations based on the *PLI* or *PLV* (post hoc *p* < 0.0001). At all included frequencies, the values of kappa were higher for the *PLI* compared with the *PLV*, while the values of ASP and mean were significantly higher for the *PLV* compared with the *PLI*. Except for in the delta band, the values of R, diameter, leaf, hierarchy, and Teff in all other bands had analogous ranges (all post hoc *p* > 0.05) whether they were computed with an application of the *PLI* or *PLV*. As shown in Figure 3, also within the HC group, there were significant differences regarding the kappa, ASP, and mean, depending on whether they were calculated based on the *PLI* or *PLV* parameters. All interaction effects (FC indicator × MST value) at the analyzed frequencies were significant at *p* < 0.0001 and *η*_p_^2^ > 0.2 after correction for multiple testing.

## 4. Discussion

The disconnection theory of schizophrenia, assuming that the disease’s etiopathogenesis is grounded in abnormal patterns of synchronizations between a given set of brain areas, is one of the major research trends in neuroscientific studies on this illness [25]. On the other hand, this conceptual orientation is strongly associated with rapidly developing methodological and computational advancements of functional connectivity (FC) and neuronal network analysis research paradigms. Although the vast majority of FC research uses fMRI and assesses levels of BOLD co-activation between brain regions occurring in common time intervals [26], the number of FC studies using EEG and MEG is still growing [27]. EEG-related FC measures can be, with some simplification allowed, divided into those based on the amplitude of the signal and those based on the phase of oscillatory activity. In general, amplitude correlations give FC arrangements similar to those obtained from fMRI; however, these indexes suffer from susceptibility to volume conduction disturbances. One way to circumvent this limitation is to design FC indicators that rely on correlations between the ongoing voltage fluctuations with time delay [28]. Due to problems related to the influence of volume conduction and signal noise on FC parameters, more and more computational methods were developed to estimate the strengths of the relationships between distinct aspects of the EEG signals and the interactions between the reconstructed signal sources. For example, Wang and coworkers [29] indicated that there are about 42 methods, while Bakhshayesh et al. [30] showed that 26 types of FC algorithms might be applied to analyze synchronizations between non-stationary, non-linear signals, such as the signal coming from EEG recordings. Unfortunately, only recently have research and computational analyses begun to evaluate which of these methods give rise to relatively repetitive connection patterns, closely related to phenotypic characteristics, that show significant interindividual and modest intraindividual variance [28,31,32]. 

Despite some advances in assessing FC measures in terms of their validity and reproducibility, according to our knowledge, the recognition of how different FC indexes may affect the scope and direction of differences between schizophrenia patients and healthy controls regarding the parameters of the global neuronal network configuration described in the language of graph theory had not yet been carried out. The answer to this question seems important, taking into account the significant range of study outcomes regarding the specificity of the organization of neural networks in schizophrenia and the observed heterogeneity of results in this area [7]. This heterogeneity was even observed within a set of studies using the same computational solutions to obtain global neural network arrangement typology, such as a minimum spanning tree [15]. In an extensive review that aimed to find a repeatable constellation of graph-theory network reconstructions based on MST, the authors indicated that, regarding the adult psychiatric population, especially those suffering from schizophrenia, the results are indisputably conflicting. However, the problem of using FC input gained from different connectivity computations was not considered as a possible important source of inconsistency [33]. To verify if two different types of FC indicators might generate different or even contradictory results for the extent that the global neural networks of schizophrenia patients differ from healthy controls, we conducted two identical computational analyses, one with the *PLI* as a synchronization measure and another with the *PLV*.

These two algorithms belong to the same group of methods assessing FC strength, i.e., those based on phase lag. According to Li et al. [34], the *PLV* is not fully resistant to the volume conduction problem, leading to an increased number of spurious synchronizations coded as genuine connections, while *PLI*, although alleviating this limitation by excluding zero-phase diversity, is less effective in managing resistance against noise. On the other hand, Rizkallah with coworkers [35] documented that the network created on the basis of the *PLV* used as an FC measure applied to EEG recordings was more similar to the fMRI network arrangement compared to a *PLI*-based network. Our results indicated that MST measures based on *PLV* indexes of synchronization differentiated schizophrenia patients from controls almost twice as often as in the case of the *PLI*. The power of group differentiation applied primarily to the index ‘Mean’ based on the *PLV*. Consequently, at all frequencies, this value was higher in patients, which suggests that their network is built with edges of higher overall weight, sometimes interpreted as the cost (e.g., the “energy”) needed to transfer the information between nodes or the length of the average edge [22].

In our opinion, the most important result of the current study is that if the groups were significantly differentiated by the same MST parameter calculated on the basis of *PLI* and *PLV*, then the result of this comparison was the opposite, i.e., the given MST parameter was higher in the clinical group compared to the healthy controls and, in the case of calculating the network marker with another FC index, was lower in the clinical group. Such results apply to global network parameters in the gamma band, especially with regard to the kappa, diameter, ASP, and mean. This seems to suggest that the outcomes of whole-brain network analyses in schizophrenia are susceptible to applied FC indicators, and depending on the applied synchronization measures, completely different values can be obtained, informing about the extent to which the organization of the neural network of schizophrenia patients differs from that of controls. None of the MST indicators showed consistently similar differences between patients and controls, whether they were calculated using connectivity strength values computed using the *PLI* or *PLV*. The inconsistency of some MST indicators, understood here as their dependence on the included FC measures, was also seen regarding intrasubject comparisons. In detail, it appeared that indexes such as the kappa, ASP, and mean were significantly different within the patient and control groups, depending on whether they were computed using the *PLI* or *PLV* indexes of functional connectivity. It is worth noting that the mean and kappa were also inconsistent regarding intergroup comparisons.

In summary, our results indicate that MST-based measures of whole-brain network arrangement encapsulated in the language of graph-theory are susceptible to the entered indexes of functional connectivity, and at least in the case of comparing patients and controls, contradictory MST values might be obtained, depending on the exact FC algorithm that is used. The above finding seems to be particularly important for schizophrenia research, on one hand because the network and connectivity analyses fit into the current understanding of the disease as a consequence of neuronal disconnection. On the other hand, the majority of already classic, yet still valid conceptual propositions suggested that schizophrenia is a disorder resulting not from pathology limited to narrowed, strictly localized cerebral dysfunction but from a general, systemic brain disorder that to varying degrees affects various areas and mechanisms, granting their mutual modulations [36,37].

Despite these important settlements, some limitations of our study and a proposal for further investigations should be addressed. First of all, it might be tentatively stated that the outcomes presented are specific to schizophrenia. It is possible that the majority of the detailed results are secondary to applied computational solutions and the analyzed discrepancies regarding MST results dependent on the included FC indexes would also appear when another clinical group is compared with healthy controls. Secondly, our study did not allow us to recognize whether MST graph-theory network characteristics are more or less susceptible to various FC measures compared with more conventional approaches analyzing such network features as small-worldness and the clustering coefficient [15,38]. Therefore, our analysis should not be considered as a clear indication of what type of network organization measures to choose in future research but rather as a study showing that the results of graph-theory network analyses depend on the built-in FC parameters. Nevertheless, some clear conclusions seemed to emerge. We postulate that it is necessary to critically evaluate these network arrangement computational indicators, which show marked intrasubject variability or are prone to exhibit such variability when even modest input design conditions are entered. The mathematically correct development of a given algorithm does not necessarily mean that it accurately depicts some aspect of the functioning of the living brain. Additionally, it seems indispensable to conduct a large-scale analysis of many FC and neuronal network indicators with various mathematical and theoretical backgrounds and verify which of them bring similar results, e.g., in the form of between-group differences, and which simultaneously have minimized intrasubject variance. It may also result that future research on disruptions in whole-brain organization in diseases such as schizophrenia will gain a greater level of reproducibility and consistency when more effort is invested in the critical analysis and selection of the already available computational possibilities in terms of their accuracy and reliability than on the further multiplication of mathematical solutions without verifying their compliance with the specificity of the activity of the brain.

## 5. Conclusions

The conducted analyzes showed that the selection of functional connectivity measures has a fundamental impact on the final results of global neural network modeling. These conclusions should be taken into account when planning research using computational analyzes of whole-brain neural networks.

## Figures and Tables

**Figure 1 brainsci-13-00138-f001:**
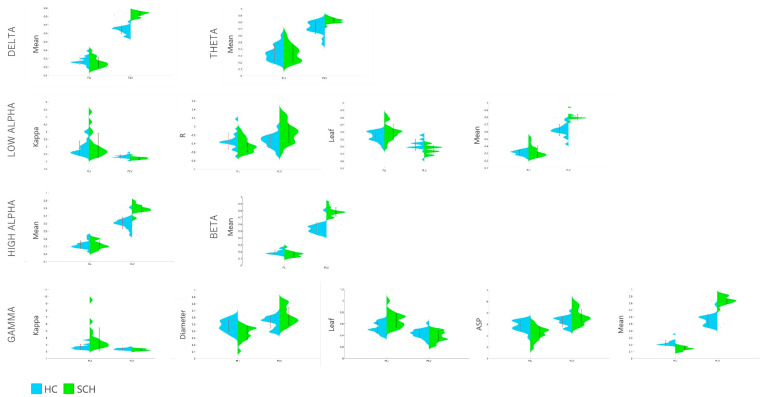
Distribution plots of selected MST indicators based on the *PLI* and *PLV* compared between groups. Blue—HC; Green—SCH. Red marks refer to means ± standard deviations. Boxplots represent only comparisons that reached the level of statistically significant differences after also applying the false discovery rate (FDR) correction for multiple testing.

**Figure 2 brainsci-13-00138-f002:**
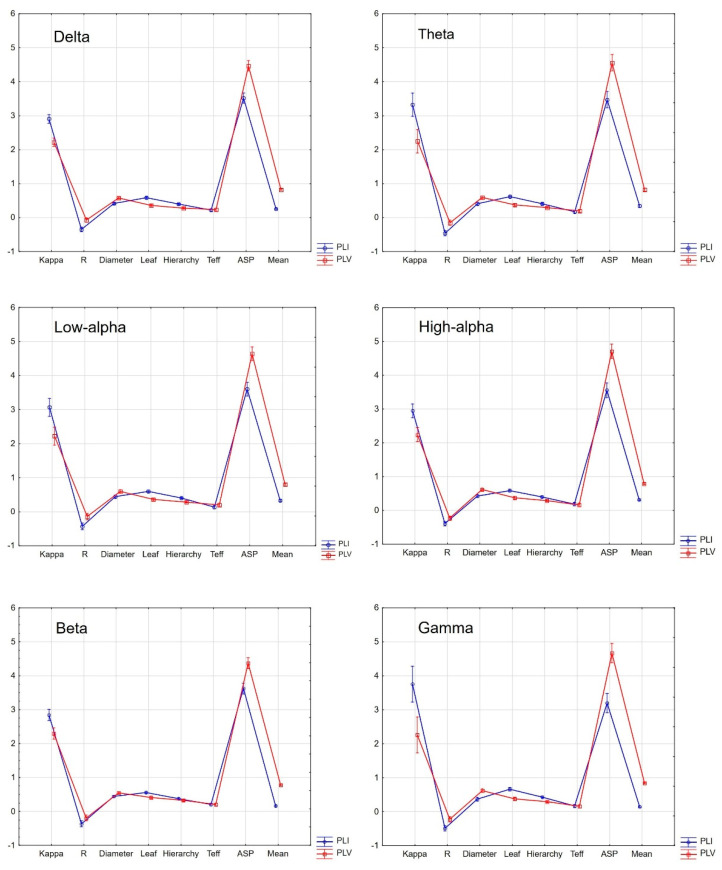
Within-group comparisons of MST indicators (kappa, R, diameter, leaf, hierarchy, Teff, ASP, and mean) based on *PLI* (blue curve) and *PLV* (red curve) FC strength indicators in the SCH group. The vertical bars represent 0.95 confidence intervals. Comparisons were selectively computed for individual frequencies.

**Figure 3 brainsci-13-00138-f003:**
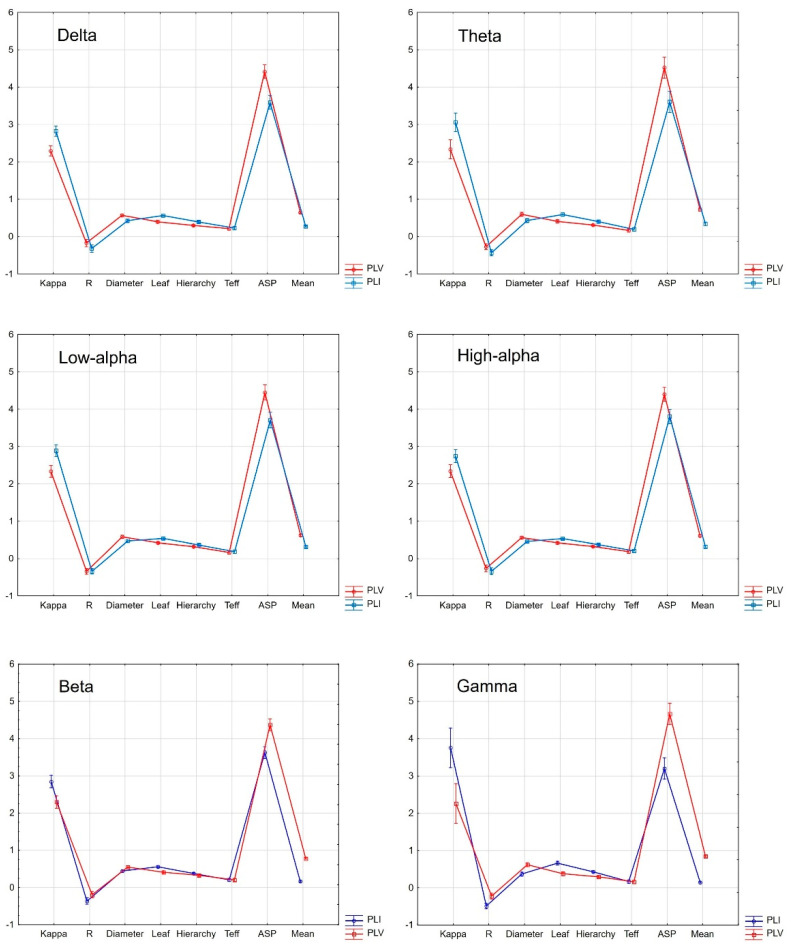
Within-group comparisons of MST indicators (kappa, R, diameter, leaf, hierarchy, Teff, ASP, and mean) based on *PLI* (blue curve) and *PLV* (red curve) FC strength indicators in the HC group. The vertical bars represent 0.95 confidence intervals. Comparisons were selectively computed for individual frequencies.

**Table 1 brainsci-13-00138-t001:** Demographic and clinical characteristics of the studied groups.

	HC *n* = 20 M (SD)	SCH *n* = 20 M (SD)	*p*
Age (years)	21.10 (1.80)	21.20 (1.96)	0.867 ^a^
Male/female	10/10	10/10	0.999 ^b^
Education (years)	13.10 (1.29)	12.90 (1.16)	0.610 ^a^
Duration of illness (months)		19.50 (4.96)	
Duration of untreated psychosis (months)		5.60 (3.10)	
Risperidone equivalents		4.82 (0.93)	
PANSS ^c^ positive subscale		14.20 (3.52)	
PANSS negative subscale		17.40 (4.89)	
PANSS general subscale		32.60 (10.93)	
PANSS total		64.20 (10.18)	

Note. ^a^ two-tailed *t* test, ^b^
*χ*^2^ test (df = 1), ^c^ Positive and Negative Syndrome Scale [24].

## Data Availability

The data that support the findings of this study are available from the corresponding author upon reasonable request.

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
