# Peer review of "How Functional Connectivity Measures Affect the Outcomes of Global Neuronal Network Characteristics in Patients with Schizophrenia Compared to Healthy Controls"

_brainsci, 2023, doi:10.3390/brainsci13010138_

Round 1

Reviewer 1 Report

This study focuses on the inconsistency of results for brain connectivity analyses in schizophrenia patients, showing that the results of graph-theory network analyses depend on the built-in functional connection parameters. The authors evaluated a wide-acknowledged but less quantitatively described problem in the area with their data. The significance of this study is just like ringing a bell to the researchers.

Section 2.3 Line 160 and Line 181. I wouldn’t call PLI and PLV “algorithms”. They are just different metrics for measuring the functional connectivity strength between nodes. You created graphs based on either of these two metrics, and then characterized the topology of the graphs with some other metrics.

Page 4, Line 191. Rewrite the formula in a correct format.

Section 3.2, Table 2. I don’t think this is a good way to present the results. Try to make graphs from these numbers and indicate the groups with significant difference, rather than showing all the raw numbers in such a table. Instead, you can attach this table at the end of the manuscript as an appendix.

Page 9, Line 306, Line 311, Line 313, Line 322. Please be consistent with the words “within-subjects” and “within-group” to avoid misunderstanding.

Figure 1 and Section 3.3. The results of within-Health group comparisons should also be shown.

Minor

Page 3, Line 146. “Darning” should be “During”. “the well-lit and quiet room” should be “a well-lit and quiet room”.

Page 4, Line 197. Delete “otherwise, it is or zero”.

Page 6, Line 276. “Differed” should be “differ”.

Figure 1. Please put the name of the frequency band (“Delta”, “Alpha”, …) above each subplot as figure titles.

Author Response

This study focuses on the inconsistency of results for brain connectivity analyses in schizophrenia patients, showing that the results of graph-theory network analyses depend on the built-in functional connection parameters. The authors evaluated a wide-acknowledged but less quantitatively described problem in the area with their data. The significance of this study is just like ringing a bell to the researchers.

  • We would like to thank the Reviewer for appreciating our manuscript.

Section 2.3 Line 160 and Line 181. I wouldn’t call PLI and PLV “algorithms”. They are just different metrics for measuring the functional connectivity strength between nodes. You created graphs based on either of these two metrics, and then characterized the topology of the graphs with some other metrics.

  • According to this suggestion, we have changed the word “algorithms” to “metrics” and
    we reworked selected sentences so that they sound stylistically correct after this correction.

Page 4, Line 191. Rewrite the formula in the correct format.

  • We have corrected the formula according to expected mathematic standard.

Section 3.2, Table 2. I don’t think this is a good way to present the results. Try to make graphs from these numbers and indicate the groups with significant difference, rather than showing all the raw numbers in such a table. Instead, you can attach this table at the end of the manuscript as an appendix.

  • We understand that a large table with a large number of quantitative results is perhaps not the most attractive form of presenting the study outcomes, however, after many efforts, we can conclude that the results contained in this table cannot be shown in the form of graph figures.

This is due to the fact that the table shows only the parameters of the MST global network, which in most cases are calculated from complex formulas and are almost exclusively numerical. Visual graphs that can be generated taking into account the MST specificity contain information based mainly on the properties of the nodes. These are the properties related to the level of centrality (e.g. betweenness centrality, BC), which we did not analyze, because the BC parameters are just variables describing more local properties of the network. The variables we analyze, describing the global dimension of the organization of the entire network, are often abstract and purely mathematical concepts, they do not have a direct graphic representation that could be read from the spanning tree. We were also unable to find in the BrainWave software used to calculate the MST network variables, parts of the software that would enable the generation of graph figures based on variables such as R, Kappa, Hierarchy, ASP, etc.

Besides, even if we managed to encode these variables somehow in a visual form, then it would require an advanced description, the legend for graphs would be a complex and quite abstract, so in the end, this could ultimately prove to be very inaccessible to readers.

Therefore, we suggest that the list of numerical results in the table is much more readable and clearly allows the reader to become familiar with the differences in MST parameters calculated on the basis of PLI or PLV.

Considering the above, we kindly ask the Reviewer to let us leave Table 2 as the main form of presenting the results.

Page 9, Line 306, Line 311, Line 313, Line 322. Please be consistent with the words “within-subjects” and “within-group” to avoid misunderstanding.

  • According to this suggestion, we unified these phrases to the form of “within-group”.

Figure 1 and Section 3.3. The results of within-Health group comparisons should also be shown.

  • According to this suggestion, we have added new Figure showing with-group comparison of MST metrics in healthy controls group (now, it is Figure 2). Due to the addition of a new figure, we have also supplemented subsection 3.3. "Within-group comparisons of PLI and PLV-based MST indicators", so that it adequately refers to the content presented in Figure 1 and added Figure 2.

Minor

Page 3, Line 146. “Darning” should be “During”. “the well-lit and quiet room” should be “a well-lit and quiet room”.

Page 4, Line 197. Delete “otherwise, it is or zero”.

Page 6, Line 276. “Differed” should be “differ”.

  • All the above errors were corrected, and the entire manuscript has undergone additional linguistic proofreading.

Figure 1. Please put the name of the frequency band (“Delta”, “Alpha”, …) above each subplot as figure titles.

  • According to this suggestion, names of the frequency band were added above each subplot in Figure 1 and Figure 2.

Reviewer 2 Report

This is a useful paper that should generate further research in the etiology of schizophrenia. The methods are sound and analysis of data is conducted by experts in the field. My suggestion is to improve the paper for the reader by having a thorough reading and correction by someone proficient in English. There are many errors in spelling and grammar including improper use of dashes, commas and "run on" sentences that are confusing. Otherwise the paper is a good report. Discrepancy in commonly used analytic tools in EEG evaluation of patients with schizophrenia should be resolved.

Author Response

This is a useful paper that should generate further research in the etiology of schizophrenia. The methods are sound and analysis of data is conducted by experts in the field. My suggestion is to improve the paper for the reader by having a thorough reading and correction by someone proficient in English. There are many errors in spelling and grammar including improper use of dashes, commas and "run on" sentences that are confusing. Otherwise the paper is a good report. Discrepancy in commonly used analytic tools in EEG evaluation of patients with schizophrenia should be resolved.

  • We would like to thank the Reviewer for appreciating our manuscript. According to the suggestion, the English-language expert has thoroughly corrected the manuscript. Many changes have been made.

Round 2

Reviewer 1 Report

For the visualization of Table 2, there are many ways to achieve, please spend more time to think about. Attached please find an example figure I made. You can use boxplots (violin plots as well) to show the distribution of your data, making your results more plausible. You can put the individual data points on top of the box plots too, like this example (https://seaborn.pydata.org/examples/horizontal_boxplot.html). Try to show more information in your data, avoid the simplest bar plots. For each power band, you can pick 2 or 3 metrics to visualize, no need to plot all of them, and indicate the statistical significance with asterisks. If you have such figures done, then you can get rid of the long table in the main text, and put it in the appendix.

Please check the entire manuscript about the spelling of some words carefully. You were right with words like “normalized”, “organization”, “conceptualized”, “synchronization”, “visualization” in the previous version. However, all of them are incorrectly spelt in the revised version, I’m wondering why.

Figure 1 and Figure 2, you put the names of the frequency bands in each subplot, which is good. However, the resolutions of all figures are really poor. Please check if there’s anything you have messed up. Moreover, even after the resolution is improved, you still need to increase the font sizes of the axis-tick labels and the axis names. Since the legends for PLV and PLI are consistent across all subplots, you only need to put a single legend at the top of the figure and delete those in each subplot.

Author Response

For the visualization of Table 2, there are many ways to achieve, please spend more time to think about. Attached please find an example figure I made. You can use boxplots (violin plots as well) to show the distribution of your data, making your results more plausible. You can put the individual data points on top of the box plots too, like this example (https://seaborn.pydata.org/examples/horizontal_boxplot.html). Try to show more information in your data, avoid the simplest bar plots. For each power band, you can pick 2 or 3 metrics to visualize, no need to plot all of them, and indicate the statistical significance with asterisks. If you have such figures done, then you can get rid of the long table in the main text, and put it in the appendix.

- According to this suggestion, we have added new Figures (2, 3) with distribution plots regarding PLI/PLV values reaching statistical significance. We have decided that these Figures will only present variables whose comparison turned out to be statistically significant because adding graphs for all possible variables and comparisons would result in the need to present nearly 100 graphs, which would not allow for their meaningful presentation at a readable level of graphic quality. Besides, it seems to us incorrect to present the same data twice.
The table with all the details was, as suggested by the reviewer, moved to the end of the text as Appendix 1.

Please check the entire manuscript about the spelling of some words carefully. You were right with words like “normalized”, “organization”, “conceptualized”, “synchronization”, “visualization” in the previous version. However, all of them are incorrectly spelt in the revised version, I’m wondering why.

- The change in the spelling of these words resulted from the choice of the English language style (previously it was British English), finally, we carried out the spelling correction in accordance with the American style, hence the return to the "organization" type of spelling.

Figure 1 and Figure 2, you put the names of the frequency bands in each subplot, which is good. However, the resolutions of all figures are really poor. Please check if there’s anything you have messed up. Moreover, even after the resolution is improved, you still need to increase the font sizes of the axis-tick labels and the axis names. Since the legends for PLV and PLI are consistent across all subplots, you only need to put a single legend at the top of the figure and delete those in each subplot.

- We have made corrected versions of the Figures with an enlarged font. Regarding the resolution of these Figures, we would like to mention that in the manuscript for review there are Figures pasted and adapted to the editing capabilities of Word. The Editorial Board has received from us separate files with high-resolution Figures, which can be inserted into the final version of the article by the technical editor of the Editorial Office. The quality of Figures pasted into a draft manuscript for review does not reflect the quality of Figures submitted outside of the manuscript itself.